# Covalently Immobilized Battacin Lipopeptide Gels with Activity against Bacterial Biofilms

**DOI:** 10.3390/molecules25245945

**Published:** 2020-12-15

**Authors:** Gayan Heruka De Zoysa, Kelvin Wang, Jun Lu, Yacine Hemar, Vijayalekshmi Sarojini

**Affiliations:** 1School of Chemical Sciences and the Centre for Green Chemical Science, The University of Auckland, Auckland 1142, New Zealand; heru.de-zoysa@auckland.ac.nz; 2School of Science, Auckland University of Technology, 34 St. Paul Street, Auckland 1142, New Zealand; kelvin.wang@aut.ac.nz (K.W.); jun.lu@aut.ac.nz (J.L.); 3Department of Biotechnology and Food Engineering, Guangdong Technion Israel Institute of Technology, Shantou 515063, China; yacine.hemar@gtiit.edu.cn; 4The MacDiarmid Institute for Advanced Materials and Nanotechnology, Wellington 6140, New Zealand

**Keywords:** antimicrobial lipopeptides, antimicrobial hydrogels, bacterial biofilm inhibition, non haemolytic, non-cytotoxic

## Abstract

Novel antibiotic treatments are in increasing demand to tackle life-threatening infections from bacterial pathogens. In this study, we report the use of a potent battacin lipopeptide as an antimicrobial gel to inhibit planktonic and mature biofilms of *S. aureus* and *P. aeruginosa*. The antimicrobial gels were made by covalently linking the *N*-terminal cysteine containing lipopeptide (GZ3.163) onto the polyethylene glycol polymer matrix and initiating gelation using thiol-ene click chemistry. The gels were prepared both in methanol and in water and were characterised using rheology, Fourier transform infrared (FT-IR) spectroscopy and scanning electron microscopy (SEM). Antibacterial and antibiofilm analyses revealed that the gels prepared in methanol have better antibacterial and antibiofilm activity. Additionally, a minimum peptide content of 0.5 wt% (relative to polymer content) is required to successfully inhibit the planktonic bacterial growth and disperse mature biofilms of *P. aeruginosa* and *S. aureus*. The antibacterial activity of these lipopeptide gels is mediated by a contact kill mechanism of action. The gels are non-haemolytic against mouse red blood cells and are non-cytotoxic against human dermal fibroblasts. Findings from this study show that battacin lipopeptide gels have the potential to be developed as novel topical antibacterial agents to combat skin infections, particularly caused by *S. aureus*.

## 1. Introduction

Antimicrobial resistance is one of the biggest health risks faced by modern society. As a result, we are increasingly reliant on “last-resort” antibiotics as “first-line” treatments against multi-drug resistant (MDR) pathogens. To further exacerbate this problem, several of these MDR pathogens also aggregate and colonise surfaces, forming bacterial biofilms that are up to 1000 times more resistant to antibiotic treatments than their planktonic counterparts [1]. An indirect effect of antibiotic resistance is the greater financial burden on patients, due to prolonged hospitalisation periods and the necessity to rely on more expensive antibiotic treatments. According to the 2013 report published by the Centre for Disease Control and Preventions each year in the United States of America, at least 2 million people become infected with resistant bacterial pathogens and at least 23,000 people die as a direct result of MDR bacterial infections [2].

Antimicrobial peptides (AMP), particularly lipopeptides, have been gaining attention as attractive alternatives to antibiotics, mainly due to their unique membrane lytic mechanism of action that reduces the chances of bacterial resistance and their ability to inhibit MDR pathogens and bacterial biofilms [3,4,5,6,7]. Battacin is a recently discovered cyclic lipopeptide with potent in vitro and in vivo antibacterial activity against MDR *P. aeruginosa* and *E. coli* [8]. Through extensive structure-activity relationship studies we have previously reported on several potent membrane-lytic linear battacin analogues with broad spectrum antibacterial activity against Gram negative (*P. aeruginosa*, *E. coli*, *E. amylovora and Pseudomoas syringae* pv. *actinidae*) and Gram positive (*S. aureus*) bacteria and fungi (*C. albicans*) [9,10,11,12,13]. Unlike the original cyclic lipopeptide, the synthetic analogues also have the ability to prevent biofilm formation and to disrupt mature biofilms of the above pathogens [9,13]. Some of our reported lipoeptide analogues are non-haemolytic (>1000 µM) against mouse red blood cells [9,13], non-cytotoxic against human dermal fibroblasts [10] and stable up to 24 h against serum proteases [12].

We have also immobilised these lipopeptides onto titanium and silicon surfaces, commonly used in medical implants without compromising their antibacterial and anti-biofilm activity against *P. aeruginosa* and *E. coli* [11]. This encouraging result prompted us to evaluate the potential of these lipopeptides as antimicrobial hydrogels. Hydrogels are three dimensional biomaterials that retain water and are extensively utilised in different medical applications such as in topical antibiotics and for wound dressing. Due to their high-water content, hydrogels provide a moisturized environment to stimulate wound healing. Additionally, the soft consistency of the hydrogel can mimic natural living tissues and prevent secondary infections caused by the entry of microorganisms into the wound [14]. Furthermore, delivery of antimicrobial compounds in the form of hydrogels can provide higher localised concentration of the biocide at the source of infection causing reduced systematic toxicity, and preventing chances of bacterial resistance [14].

Hydrogels can be produced by either self-assembly of molecules, dictated by non-covalent interactions such as hydrophobic forces and π–π stacking or covalent crosslinking using different polymers (e.g., chitosan, alginate and poly(ethylene glycol) methacrylate) [15,16,17,18,19]. We recently reported on ultrashort battacin based lipopeptides that form self-assembling antimicrobial hydrogels with activity against *P. aeruginosa* and *S. aureus* [10]. One drawback of self-assembling hydrogels is the requirement for high biocide concentration, which makes them an expensive option for drug delivery applications [14,18]. Additionally, non-covalent interactions which trap the self-assembled biocides, can facilitate rapid drug release leading to reduced efficacy. Consequently, antimicrobial compounds with toxicity at higher doses and chances of developing antibacterial resistance are not ideal candidates for self-assembling antibacterial hydrogels [14,18,19]. Covalently assembled hydrogels can mitigate these issues. Covalent linkage can precisely control the amount of the antimicrobial agent loaded onto the hydrogel matrix, which reduces systemic toxicity and prevent any biocide leakage from the hydrogel scaffold, leading to longer-term efficiency. This study reports the synthesis, characterisation, and antimicrobial activity of covalently conjugated linear battacin lipopeptide based hydrogels.

## 2. Results and Discussion

### 2.1. Hydrogel Design

We have previously immobilised the *N*-terminal cysteine containing lipopeptide (GZ3.163) onto glass, silicon, and titanium surfaces using polyethylene glycol (PEG) as a linker [11]. PEG based antimicrobial hydrogels have been extensively studied in the literature and PEG based materials are ideal for antibacterial surfaces mainly due to their anti-adhesive property attributed to the high mobility and steric hindrance of the ethylene glycol moieties as well as due to low host immune response [14,19,20,21,22,23]. The hydrogels reported in this study were generated using PEG based polymer as the hydrogel matrix and thiol-ene click chemistry to covalently link the *N*-term cys lipopeptide to the hydrogel network (Scheme 1). Thiol-ene click reaction is highly chemo-selective, has fast reaction kinetics, requires mild reaction conditions, can be carried out in inert solvents such as water, and high yields of pure products without the need for chromatographic separation can be achieved [24].

The lipopeptide containing hydrogel (0–10 wt% to the total polymer content) was prepared by mixing the tetra branched thiol crosslinker; PTMP diacrylate spacer, PEGDA, and a catalytic amount of the photoinitiator DMPA. Controlled hydrogel formation was achieved via photopolymerization under UV irradiation to generate reactive radicals, which undergo selective Michael reaction between thiol and acrylate groups resulting in irreversible thioether bond between the lipopeptide and PEG matrix [20,21]. Several groups have utilised this unique thiol-ene photopolymerization reaction for a range of biomedical applications [20,21,22,23,25,26]. To the best of our knowledge, only Cleophas et al. have covalently immobilised antimicrobial peptides onto the PEG hydrogel matrix using this chemistry (Scheme 1) [20,21]. In their work, the *N*-terminal cys containing antimicrobial peptide, HHC10 (0.1%, 0.5%, 1% and 10 wt% to the total polymer content) was covalently immobilised onto the hydrogel network using thiol-ene photopolymerization [21]. The antibacterial activity of these hydrogels was evaluated against *S. aureus*, *S. epidermidis*, and *E. coli*. Only hydrogels carrying the 10 wt% of the peptide was efficient in eradicating these pathogens [21]. Although the HHC10 peptide is non-haemolytic against the sheep blood cells, the haemolytic activity and cell viability of the peptide conjugated hydrogels have not been reported by the authors [20,21].

As evident from the Scheme 1 images (before and after polymerization), GZ3.163 peptide-gels at 0.1, 0.5, 1, and 10 peptide wt% was successfully generated using thiol-ene photopolymerization. The PEG hydrogel without the peptide (i.e., 0% peptide weight) was used as the control sample. All samples had very fast gelation kinetics (<60 s) except for the sample containing 10% wt. of the peptide, which took 10 min to gelate. The gel samples were initially prepared in methanol as reported by Cleophas et al. [20,21]. We decided to generate these peptide hydrogels (0, 0.1, 0.5, and 1 peptide wt% to total polymer content) in water as well in order to more closely mimic physiological conditions. Although GZ3.163 is highly soluble in water, both the PEG spacer and the tetra branched thiol crosslinker are highly insoluble in water, hence, a milky solution resulted upon mixing them together before attempting the photopolymerization (Appendix A). Regardless, GZ3.163 did form covalently linked hydrogels in water (Appendix A) with similar reaction kinetics to the methanol samples.

### 2.2. Characteristics of the Hydrogel

#### 2.2.1. Rheology

Peptide hydrogel formation was confirmed by the tilt test (Scheme 1 hydrogel image after polymerization and Appendix A). Mechanical properties of the peptide-PEG gels in methanol and water were further analysed using rheology (Figure 1 and Appendix A).

As evident from Figure 1, the storage modulus of all the gels were significantly greater than the loss modulus, indicating the formation of an elastic gel network rather than a viscous liquid. The presence of the lipopeptide at varying concentrations did not affect gel formation in any way. However, the intensity of the storage and loss moduli between the methanol and water samples were different, possibly due to solubility issues of the PEG spacer and the crosslinker in water. Furthermore, all of the gel samples showed significant decrease in viscosity with increasing sheer rate (Appendix A) similar to gels reported by Lee et al. [27]. These gels exhibit sheer thinning ability as the physical linkage of the PEG matrix breaks upon applying shear stress. These are desirable characteristics for topical antibacterial applications, injectable gels and eye drops.

#### 2.2.2. Fourier Transform Infrared (FT-IR) Spectroscopy

Successful conjugation of the lipopeptide onto the gel matrix was confirmed by FT-IR spectroscopy (Figure 2). With 10 wt% lipopeptide in methanol, a peak at 1668 cm^−1^ was observed. This peak corresponds to the amide I region and is mainly associated with carbonyl vibrations arising from the peptide molecules. The 10 wt% lipopeptide in water also observed a peak at 1641 cm^−1^ corresponding to amide I region (Appendix A). This characteristic peak was absent in the control gel sample devoid of the lipopeptide (0 wt% lipopeptide.) At lower lipopeptide concentrations (0.1–1 wt%) the gels did not show the characteristic amide I and II bands possibly due to reduced lipopeptide concentration to yield any detectable signal [28].

#### 2.2.3. Scanning Electron Microscopy (SEM)

The morphology of the lipopeptide gels were studied using SEM (Figure 3). As evident from Figure 3, the gels prepared in water and methanol exhibit different surface morphologies possibly due to the insolubility of PEG spacer and the tetrabranched cross linker in water. The hydrogels prepared in water (Figure 3, left) have a smoother surface compared to the gels prepared in methanol. However, they exhibit a criss-cross pattern throughout the surface as well as granular spherical particles embedded within the surface. It is possible that these granules are the insoluble starting materials while the criss-cross architecture could be the cross-linked polymer matrix. The methanol containing gels were highly coarse, which may be due to the extensive covalent crosslinking of the PEG materials. Unlike the gels prepared in water, those prepared in methanol were porous with pore sizes ranging from 0.2 to 1.9 µm. Such porous gels could have potential applications in wound dressing, as the porous structure in these gels is important for oxygen transport and absorption of exudates. The architecture of the gel remained the same for both 0 and 1 wt%.

### 2.3. Antibacterial Analysis

The antibacterial analysis of the lipopeptide containing gels (Figure 4) were performed against *P. aeruginosa* as a model Gram negative pathogen and *S. aureus*, as one of the common bacterial pathogens associated with skin infections.

The lowest weight percent of the peptides that showed antimicrobial activity in the hydrogels reported by Cleophas et al. was 10 wt% (relative to polymer content) against *S. aureus*, *S. epidermidis* and *E. coli* [20,21]. Our lipopeptide gels showed excellent antibacterial activity at a peptide content 100 times lower (0.1 wt%) against *P. aeruginosa* and *S. aureus.* Lipopeptide gels prepared in methanol showed up to 97% bacterial growth inhibition (Figure 4a,c) against the two pathogens. In fact, no live bacteria were recovered from the samples (Figure 4b,d). These results indicate that the antibacterial activity of these lipopeptide gels are initiated upon bacterial contact (contact killing) with the gel sample. However, only 0.5 and 1 lipopeptide wt% hydrogels in water showed potent antibacterial activity against *P. aeruginosa* and *S. aureus*. The discrepancy in the antibacterial activity between the gels prepared in methanol and water could be due to poor solubility of PEG matrix in water, as described earlier. The control PEG only gel (0 wt% of the lipopeptide) showed weak antibacterial activity against the two pathogens, possibly due to the antiadhesive properties of PEG. Weak antibacterial activity of PEG coated surfaces has been observed in our previous work [11].

### 2.4. Anti-Biofilm Activity

The gel samples at 0.5 to 1 wt% in methanol showed potent antibacterial activity (92–100% growth inhibition) against the cells released from both *P. aeruginosa* and *S. aureus* biofilms (Appendix A). The lipopeptide hydrogels at 0.5 and 1 wt% prepared in water showed moderate antibacterial activity (56–60% inhibition) against *P. aeruginosa* cells released from the mature biofilm (Appendix A). However, the antibacterial activity of lipopeptide hydrogels against cells released from the *S. aureus* biofilm (Appendix A) was weak (23% inhibition at 1 wt%).

The ability of the lipopeptide gels to disrupt mature biofilms (7 days old) of *P. aeruginosa* and *S. aureus* was studied using a crystal violet staining assay. Crystal violet is known to bind to the exopolysaccharide matrix (EPS) in the biofilm. The stained biofilm can be solubilised upon addition of ethanol for a semi-quantitative estimation of the attached biofilm biomass. As evident from Figure 5a,c, hydrogel samples prepared in water did not completely eradicate (23% biomass reduction at 1 wt%) the mature biofilms of *P. aeruginosa* (Figure 5a). The 1 wt% lipopeptide in water eradicated up to 59% of the mature biofilms of *S. aureus* (Figure 5c). The 0.5 and 1 wt% lipopeptide-gels in methanol exhibit 70 to 86% reduction in biomass (Figure 5a,c) which is a promising result when it comes to eradicating mature biofilms of the two pathogens. Unlike the gel samples prepared in water, no mature biofilm cells were recovered from 0.5 wt% and 1 wt% methanol gel samples (Figure 5b,d).

These results indicate that the lipopeptide gels prepared in methanol have promising antibacterial and anti-biofilm activity against the two bacteria even though eradication of mature biofilms required slightly higher lipopeptide content (≤0.5 wt%) than required for eradicating the planktonic (≤0.1 wt%) cells. This result is not surprising as higher bacterial cell density is observed in mature biofilms. The bacterial cells within the biofilm are surrounded by EPS that accounts for 50 to 90% of the biomass in the biofilm. The EPS acts as a physical barrier against hostile conditions such as UV exposure, acid stress, and metal toxicity, thereby acting as the first line of defence against antibiotics [1,29]. Because of being covalently linked to the PEG matrix, the lipopeptide molecules are unlikely to leach out from the gel matrix and diffuse through the EPS, which would explain the need for a higher peptide content for the eradication of the mature biofilms. Nevertheless, these lipopeptide gels have the ability to prevent colonisation of biofilms onto new sites due to their contact kill mechanism of action.

### 2.5. Haemolysis of Mouse Blood Cells

To qualify for clinical use, toxicity below a particular threshold is essential in addition to antibacterial potency. Previously we have shown that several peptides from the battacin lipopeptide library have negligible haemolytic activity against mouse blood cells [9,11,13]. It was still important to determine the haemolytic activity of the lipopeptide gels against mouse blood cells to rule out that this formulation is not haemolytic (Figure 6) as our self-assembling gels were haemolytic.

Both the gels prepared in methanol and water in the absence of the lipopeptide (0 wt%) showed negligible (9–11%) haemolysis against mouse blood cells. In the presence of the lipopeptide, a slight increase in haemolysis (5–20%) was observed but was within the expected range (≥20%) for peptide hydrogels reported in the literature [30,31,32,33].

### 2.6. In Vitro Cytotoxicity Against Human Dermal Fibroblast Cells

Our ultrashort battacin based lipopeptides exhibited a 50% cell viability at ≥125 µM between a 24 and 72 h time period [10]. The cytotoxic activity of synthetic battacin lipopeptide analogue GZ3.163 has not been previously investigated. The cytotoxicity of GZ3.163 was evaluated following a MTT cell viability assay of human dermal fibroblasts (Appendix A). As evident from Appendix A, up to 50% cell viability was observed for GZ3.163 at ≥125 μM between a 24 to 72 h time period. The cytotoxicity range observed is same as the ultrashort battacin lipopeptides [10] and similar to what has been reported previously for the natural product battacin (128 µM against the HEK 293 human kidney cell lines) [8].

The results from the cell viability assay on the lipopeptide gels are shown in Figure 7. In general, cell viability was found to increase with increasing concentration of the peptides in the gels. It was observed that increased contact time had a detrimental effect on cell viability, which changed from 109 to 140% at 24 h to 75 to 86% in 72 h (Figure 7). The fact that >75% cell viability was retained after 72 h of contact with the lipopeptide gels prepared in both methanol and water is very encouraging and prove the non-cytotoxic nature of these lipopeptide gels against the human dermal fibroblasts, which is within the expected range (60–100%) for peptide hydrogels reported in the literature [14,22,27,30].

## 3. Conclusions

Thiol-ene click chemistry was used to convert a linear lipopeptide from the battacin family into an antimicrobial gel by covalent conjugation to PEG matrix. These lipopetide gels were characterised by rheology and successful immobilisation of the peptide component in the gel network was confirmed using FTIR, which showed a dominant peak at 1668 cm^−1^ attributed to the amide I band of the peptide molecules. The gel containing as low as 0.1 wt% of the lipopeptide was capable of completely inhibiting the growth of *P. aeruginosa* and *S. aureus*, which is a significant improvement to previous literature reports where bacterial growth inhibition could not be achieved below 10 wt% of the peptide component. None of the previous reports have investigated biofilm inhibition using antimicrobial peptide gels or have presented any toxicity data. Our antibacterial and anti-biofilm analyses have shown that at a minimum peptide content of 0.5 wt%. the gels prepared in methanol, successfully inhibit both planktonic and mature biofilms of *P. aeruginosa* and *S. aureus*. Similar to our previously reported lipopeptides from this family, these synthetic peptide gels were not haemolytic against mouse blood cells. The control gels prepared in methanol showed negligible haemolytic activity and cytotoxicity against human dermal fibroblasts, indicating that the chemistry used here would be amenable for developing drug formulations that can be directly used in patients to treat infections. Additionally, incorporation of the peptide into the PEG gel matrix lead to an increase in cell viability, which implies that the peptide gel provides favourable conditions for cell harvesting. In summary, results from the current investigations prove that the battacin lipopeptide gels are ideal candidates that hold promise for development as novel topical antibiotics because of their antibacterial and antibiofilm activities and promising safety profile.

## 4. Materials and Methods

### 4.1. Chemicals and Reagents

All solvents were of analytical grade and were used without further purification. Pentaerythritol tetrakis(3-mercaptopropionate) (PTMP), poly(ethylene glycol) diacrylate (PEGDA) (Mn ~700), 2,2-dimethoxy-2-phenylacetophenone (DMPA), crystal violet, and 3-(4,5-dimethylthiazol-2-yl)-2,5-dipheynyltetrazolium bromide (MTT) were purchased from Sigma Aldrich (St. Louis, MO, USA). Human dermal fibroblasts (Cell line HDFa, catalogue number: C0135C) and low serum growth supplement (LSGS, Catalog number: S00310) were purchased from Life Technologies NZ (Auckland, New Zealand). The *N*-terminal cys containing lipopeptide (GZ3.163) was synthesised following standard solid phase peptide synthesis protocols using Fmoc chemistry. The detailed experimental procedure for the synthesis, purification, and characterisation of this peptide has been reported in our previous publications [9,11,13].

### 4.2. Gel Formation 

PTMP (2 g, 4.1 mmol), PEGDA (7 g, 10 mmol) and catalytic amount of DMPA as the photoinitiator (0.1 wt%, ~10 mg) were mixed together. The lipopeptide at 0.1 wt% (0.4 mg, 0.26 µmol), 0.5 wt% (2 mg, 1.3 µmol), 1 wt% (4 mg, 2.6 µmol), and 10 wt% (40 mg, 26 µmol), relative to polymer content was dissolved in either methanol (1.33 mL) or water (1.33 mL). The polymer mixture (0.33 mL, 389 mg) was added dropwise to the peptide solution. The polymer mixture was completely soluble in methanol but was insoluble in water. The gelation was initiated with a UV lamp at 365 nm (intensity ca. 4.6 mw cm^−2^ or 25 mWcm^−2^ for 15 min) for 1 to 10 min. Successful gelation was initially validated by inverting the vial (Figure 1 and Appendix A) [20,21].

### 4.3. Rheology 

Oscillatory rheometer (MCR 302, Anton Paar Austria) with a stainless-steel plate (25 mm diameter) at 25 ± 1 °C was used to further characterise these gels. A gap height of 0.2 mm was maintained between the plates before each measurement. A dynamic sweep was applied at an angular frequency of 6 rad/s. Finally, a frequency sweep of 1 to 100 rad/s with 1% strain amplitude (γ) was used to record the viscoelastic behaviour. The storage modulus (G′) and loss modulus (G′′) were calculated as a function of angular frequency at every point. For shear-thinning experiment, hydrogels were examined as a function of shear rate from 0.1 to 1 s^−1^ [18,27]

### 4.4. Fourier Transform Infrared (FT-IR) Spectroscopy 

FT-IR spectra were recorded using a PerkinElmer Spectrum Two FT-IR spectrometer (PerkinElmer, Buckinghamshire, UK) fitted with an ATR diamond crystal attachment, operated in single bounce mode. The gel samples were scanned from 400 to 4000 cm^−1^ and at every 2 cm^−1^ increments, an average of 36 scans were recorded. The spectrum from methanol was subtracted as the background [10,34].

### 4.5. Scanning Electron Microscopy 

The gel samples were thoroughly freeze dried to obtain the dry gel. The dried samples were placed in a carbon tape onto an aluminum stud and sputter coated with gold, for 20 s at 25 mA before viewing under high vacuum using Hitachi Su-70, SEM microscope (Hitachi, Tokyo, Japan), at 5 kV at the Faculty of Science, Auckland University of Technology [35].

### 4.6. Antibacterial Analysis 

*S. aureus* and *P. aeruginosa* ATCC 27853 were obtained from the microbial culture collection of the School of Biological Sciences or the Faculty of Medical and Health Sciences, at the University of Auckland. These two strains produce biofilms, as reported on our previous study [9] and by others [36]. For the antibacterial assay, a single colony of bacteria was transferred to and grown overnight in Muller Hinton broth (MHB) broth. The optical density of the overnight culture was adjusted to 0.5 Mcfarland standard and was further diluted 100 times to obtain a final bacterial concentration of 10^6^ CFU/mL. The pre prepared gel samples (25 mg ± 3 mg) in methanol and water at 0%, 0.1%, 0.5% and 1% lipopeptide weight relative to polymer content were placed in polypropylene coated 96 well plates (greiner-Bio-one, MediRay). The diluted bacterial solution (100 μL) was added to each well. For comparative purposes, the plates also had both growth and sterility controls (negative control) included, both of which did not contain the gels. The positive control was the GZ3.163 lipopetide (1 mM) which has previously shown activity against *S. aureus* and *P. aeruginosa* [11]. The plates were incubated overnight at 37 °C without any agitation. The gels were removed from the 96 well plates and washed thrice with sterile saline solution (0.9% NaCl, 200 μL) to remove any non-adhered bacteria. The optical density of the bacterial supernatant in the 96 well plate was measured at 600 nm using an EnSpire Multimode plate reader and the % of bacterial inhibition was calculated using the following equation
% bacterial growth inhibition = (1 − (Aexp − A_MHB_)/(A_GC_ − A_MHB_) × 100),
where, Aexp is the absorbance of the supernatant from the wells containing the gel samples, A_MHB_ is the absorbance of the MHB broth (sterility control) and A_GC_ is the absorbance of the bacterial growth control in the absence of gel samples.

The washed gels were transferred to fresh saline (0.5 mL) and sonicated for 5 min at 30 W to release the adhered bacteria on the gel surfaces. The retrieved bacteria were further diluted 10,000 times with saline and 100 μL of the diluted bacterial culture was plated onto MHB agar plates and incubated at 37 °C overnight. The following day, the number of colonies on each plate was counted to assess the antibacterial potency of the gels. These experiments were done in triplicate and repeated three times on independent days [11,18,20,21].

### 4.7. Anti-Biofilm Analysis 

*P. aeruginosa* and *S. aureus* cultures grown overnight were diluted to a final bacterial concentration of 10^6^ CFU/mL and 100 μL of the diluted bacterial culture was inoculated into a tissue culture treated flat bottom polystyrene 96 well plate (Corning, Sigma Aldrich) and mature biofilms were allowed to grow for seven days at 37 °C without agitation. During each day, the bacterial supernatant was carefully removed without disturbing biofilm cells at the bottom of the wells, the wells carefully were washed twice with saline (200 μL) to remove any planktonic cells and were replaced with fresh MHB medium. On the seventh day, gel samples were added to the mature biofilms and the plate incubated overnight at 37 °C. The assay included growth and the sterility controls. The gels were carefully removed without disrupting the biofilms, and were washed thrice with saline to remove any loosely adhered biofilm cells. The CFU assay to assess the potency of the gels on the biofilm cells was conducted following the same methodology described in the antibacterial section. The bacterial supernatant from each well was carefully transferred to the wells of a fresh 96 well plate and the optical density and the percentage of bacterial inhibition determined using the methodology described under antibacterial analysis.

The ability of the gel samples to disrupt mature biofilms was determined using a crystal violet staining assay. After removing the supernatant, the biofilms were gently washed thrice with saline (100 μL) to remove any planktonic cells. The biofilms were then fixed by adding methanol (100 μL) for 15 min, after removing which, the biofilms were washed once with saline. Crystal violet (1% *w*/*v*, 100 μL) was added to each well and the plate incubated for 10 min at room temperature. After removing any excess stain, the wells were washed thrice with saline. The stained biofilm was solubilised by adding 96% ethanol (100 μL) for 5 min. The solubilised dye from each well was transferred to a fresh plate and optical density at 560 nm measured for a semi quantitative estimation of the bacterial biofilm mass. The percentage reduction in bacterial biofilms was calculated using the formula described under the antibacterial analysis. The experiment was performed in triplicates and was repeated two times on independent days. [9,27,37]

### 4.8. Haemolytic Assay 

The procedure reported in our previous publications [9,11,13] was followed for the haemolytic assay, with minor modifications as described below. Freshly collected mouse blood cells were centrifuged at 1000 g for 5 min to remove the buffy coat. The blood cells were washed three times in Tris buffer (10 mM Tris, 150 mM NaCl, pH 7.4) and re-suspended in 2% (*v*/*v*) Tris buffer. The lipopeptide gels (peptide concentration ranging from 0.21 to 2.1 mM) pre prepared in methanol and water (25 mg ± 2 mg) were placed into the wells of a 96 well plate and the re-suspended blood cells (200 μL) were added to the gel samples. The Tris buffer solution and 0.1% Triton X-100 were used as the negative and positive controls respectively. The plates were incubated for 1 h, at 37 °C without agitation. The gel samples were removed and the plates were centrifuged at 3500 g for 10 min. The supernatant (100 μL) from each well was carefully transferred into a new plate and the absorbance at 540 nm measured. The experiment was done in triplicate and repeated twice on independent days. Percentage haemolysis of lipopeptide gels was determined using the following equation,
% haemolysis = (Aexp − ATris)/(A100% − ATris) × 100,
where, A_exp_ is the absorbance of the gel samples at 540 nm, A_Tris_ is the absorbance of negative control and A_100%_ is the absorbance of the 0.1% Triton X-100 solution (positive control) [9,11,13].

### 4.9. Cytotoxicity Assay 

The cytoxicity assay protocol previously reported by us was followed with minor modifications. Human dermal fibroblast (cell line HDFa), grown in medium 106 supplemented with low serum growth supplement, were seeded (100 μL) into tissue culture treated 96 well plates at a cell density of 50,000 cells/mL and incubated overnight at 37 °C with 5% CO_2_. After overnight incubation, the media was carefully removed, and the lipopeptide gel samples were placed into the respective wells. Fresh growth media was added to each well and the plates were incubated for 24, 48, and 72 h time intervals. At each time point, the gel samples and the media were carefully removed exposing the treated cells. The cell viability, reported as percentage of the dermal fibroblast cells treated with the lipopeptide gels compared to the control, was verified by MTT assay (Sigma, Auckland, NZ), carried out according to the manufacturer’s instructions. The Viable cell amount (based on the intensity of the MTT-formazan complex) was determined by measuring absorbance at 540 nm using a multiplate reader (UV-visible FLUOStar Omega Multidetection Microplate Reader, AlphaTech NZ Ltd., Auckland, New Zealand). The experiment was carried out in triplicates and repeated two times on independent days [10].

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
