# Peer review of "Covalently Immobilized Battacin Lipopeptide Gels with Activity against Bacterial Biofilms"

_molecules, 2020, doi:10.3390/molecules25245945_

Round 1
Reviewer 1 Report
Though, the manuscript is written well but needs to be checked for spelling and grammatical mistakes like misspelled words, errors with punctuation, etc throughout.
Materials and Methods
- Fourier Transform Infrared (FT-IR) Spectroscopy need reference
- How sample was prepared for Scanning Electron Microscopy descrbied in details with appropriate references (https://link.springer.com/article/10.1007/s11356-018-3212-7)
- Provide referces that the used standard ATCC strains i.e., S. aureus and P. aeruginosa ATCC 27853 were biofilm positive.
- ForAnti-biofilm Analysis cite latest referene (https://www.mdpi.com/2079-6382/9/9/572). Also mentioned the concentartion peptie used for biofilm assay
- In biofilm assay; Crystal violet (1% 165 w/v, 100 μL) was added to each well and the plate incubated for 10 minutes. What was the condion for incubation with crystal violet.
- Haemolytic Assay: concentartion of compunds is not mentioned.
- Did authors used round-bottom 96-well microtiter plates for biofilm formation assay or Flat bottom 96- well .
- For antibacterial acivity what was the positve and negative control.
- Which bacteria is more susceptible to tested peptide.
- What the mode of action of tested peptides against these bacterial strains.
- Antibiofilm assay shows that tested peptide inhibit Aeruginosa biofilm by 56-60% while the inhibition was 23% for S. aureus biofilm. Describe the reason and mechanims.
- Why methonal was used. Can we used other organic solvent
- The references quoted within the manuscript should be checked for uniformity
- Discussion needs more elaborations.
Author Response
Comments
- Fourier Transform Infrared (FT-IR) Spectroscopy need reference
Our response
Our previous paper which described the FT-IR methodology and general information about FTIR was cited line 110.
- How sample was prepared for Scanning Electron Microscopy descrbied in details with appropriate references (https://link.springer.com/article/10.1007/s11356-018-3212-7)
Our response
Please see the track changes line 117 with appropriate reference
- Provide referces that the used standard ATCC strains i.e., S. aureus and P. aeruginosa ATCC 27853 were biofilm positive.
Our response
On our previous publication, we have shown morphological evidence of both S. aureus and P. aeruginosa producing biofilms and these strains were used for the current study as well.1 Furthermore, Brown et al. also used the same strain to report P. aeruginosa biofilms.2 These references have been added to the main text (line 123-124).
- ForAnti-biofilm Analysis cite latest referene (https://www.mdpi.com/2079-6382/9/9/572). Also mentioned the concentartion peptie used for biofilm assay.
Our response
The latest reference was added line 146
- In biofilm assay; Crystal violet (1% 165 w/v, 100 μL) was added to each well and the plate incubated for 10 minutes. What was the condion for incubation with crystal violet.
Our response
The crystal violet staining was carried out in room temperature (see line 166)
- Haemolytic Assay: concentartion of compunds is not mentioned.
Our response
The peptide concentration in the gels range from 0.21-2.1 mM (line 179)
- Did authors used round-bottom 96-well microtiter plates for biofilm formation assay or Flat bottom 96- well .
Our response
We used flat bottom 96-well plates (line 150)
- For antibacterial acivity what was the positve and negative control.
Our response
The positive control was the GZ3.163 lipopeptide (1 mM) itself which has previously shown potent activity against S. aureus and P. aeruginosa3 and the negative control was the sterile broth.
- Which bacteria is more susceptible to tested peptide.
Our response
Based on our previous study GZ3.163 is slightly more active against P. aeruginosa (1.25-2.5 µM) than S. aureus (2.5-5 µM).3
- What the mode of action of tested peptides against these bacterial strains.
Our response
These peptides have a membrane lytic mechanism of action and was added in line 49.1, 3
- Antibiofilm assay shows that tested peptide inhibit Aeruginosa biofilm by 56-60% while the inhibition was 23% for aureus biofilm. Describe the reason and mechanims.
Our response
We hypothesise that the mechanism of action of peptide-gel is membrane lysis upon contact with bacteria. This has been observed in our previous study where GZ3.163 was immobilized onto titanium, glass and silicon surfaces The SEM images of P. aeruginosa in the presence of peptide immobilised surfaces showed similar morphological characteristics as the free peptide in the solution.1, 3
The difference in the activity of bacterial cells released by mature biofilms of P. aeruginosa and S. aureus for lipopeptide hydrogels prepared in water could be due to the inhomogeneous distribution of the lipopeptide in the hydrogel matrix. As evident from the SEM images (Figure 3), the peptide hydrogel in water has a smoother surface with criss-cross pattern which we believe is the cross-linked peptide polymer matrix with granules (insoluble starting materials) spread throughout the gel. However, the methanol gel is uniform with extensive crosslinking of peptide-PEG material. It is plausible that the peptide maybe deeply buried in the hydrogel sample which may have prevented the S. aureus cells released from the mature biofilm making contact with the peptide in the gel sample to initiate the membrane lytic mechanism of action
- Why methonal was used. Can we used other organic solvent
Our response
We wanted to replicate the original hydrogelation procedure reported by Cleophas et al.4, 5 who used methanol to initiate thiol-ene click photopolymerisation reaction. Ethanol can also be used to initiate the gelation procedure.
- The references quoted within the manuscript should be checked for uniformity
Our response
The references have been amended to ensure uniformity throughout the manuscript.
- Discussion needs more elaborations.
Our response
We strongly believe, our discussion is thorough as the other two reviewers did not raise any issues with this section.
References
1 De Zoysa, G. H.; Cameron, A. J.; Hegde, V. V.; Raghothama, S.; Sarojini, V., Antimicrobial peptides with potential for biofilm eradication: synthesis and structure activity relationship studies of battacin peptides. J Med Chem 2015, 58 (2), 625-39.
2 Brown, M. L.; Aldrich, H. C.; Gauthier, J. J., Relationship between Glycocalyx and Povidone-Iodine Resistance in Pseudomonas-Aeruginosa (Atcc-27853) Biofilms. Applied and Environmental Microbiology 1995, 61 (1), 187-193.
3 De Zoysa, G. H.; Sarojini, V., Feasibility Study Exploring the Potential of Novel Battacin Lipopeptides as Antimicrobial Coatings. ACS Appl Mater Interfaces 2017, 9 (2), 1373-1383.
4 Cleophas, R. T.; Sjollema, J.; Busscher, H. J.; Kruijtzer, J. A.; Liskamp, R. M., Characterization and activity of an immobilized antimicrobial peptide containing bactericidal PEG-hydrogel. Biomacromolecules 2014, 15 (9), 3390-5.
5 Cleophas, R. T. C.; Riool, M.; van Ufford, H. C. Q.; Zaat, S. A. J.; Kruijtzer, J. A. W.; Liskamp, R. M. J., Convenient Preparation of Bactericidal Hydrogels by Covalent Attachment of Stabilized Antimicrobial Peptides Using Thiol-ene Click Chemistry. Acs Macro Letters 2014, 3 (5), 477-480.
Reviewer 2 Report
The paper is interesting, however authors need to put some points in methods and results.
Is necessary to do a statistical analysis of antibacterial and antibiofilm analysis in order to complete the results discussion and the respective figures of that section.
Author Response
Comments
- Is necessary to do a statistical analysis of antibacterial and antibiofilm analysis in order to complete the results discussion and the respective figures of that section.
Our response
Please refer to the updated Figure 4 (line 322-323) and 5 (line 370-371).
Reviewer 3 Report
The manuscript by Prof. Geyan Heruka De Zoysa and colleagues entitled "Covalently immobilized battacin lipopeptide gels with activity against bacterial biofilms" reports the formation of antimicrobial gels in water and methanol by N-terminal cysteine containing lipopeptide (GZ 3.163), the characterization using rheology, FTIR, SEM. This study may be valuable for the antimicrobial applications on the gel by the peptides. I recommend a major revision of this manuscript considering the following issues.
1.
The authors formed two gel of the peptide by water and methanol. The authors mentioned the gel prepared in methanol to “hydrogel”. Using only organic solvents, the authors should mention it to “organo-gel”.
Line 287, 289 … , Figure 1 and so on.
The author only focuses on the presence of lipopeptides in the FTIR of Figure 2. The authors should consider each of the other IR bands further. And, could you show the FTIR spectra of hydrogel prepared in water?
3.
G’ and G’’ values of the gel in water were significantly different from organo gel in methanol. It may suggest that the peptide have different interactions with the solvent. Could the authors explain the reason in the body?
In figure 4-6, the antimicrobial activity was exhibited. The antibacterial activity of water gels does not show systematic changes depending on peptide concentration. I think the data is unreliable. Further, the gels prepared in methanol are fairly active, but this is still more influenced by the direct interaction of the organic solvent methanol with bacteria than the gel formation itself. Therefore, reviewers believe that the effects of gel activity are unlikely.
Round 2
Reviewer 3 Report
 _